# Genetic Parameters and Genome-Wide Association Studies for Anti-Müllerian Hormone Levels and Antral Follicle Populations Measured After Estrus Synchronization in Nellore Cattle

**DOI:** 10.3390/ani10071185

**Published:** 2020-07-13

**Authors:** Laís Grigoletto, Miguel Henrique Almeida Santana, Fabiana Fernandes Bressan, Joanir Pereira Eler, Marcelo Fábio Gouveia Nogueira, Haja N. Kadarmideen, Pietro Sampaio Baruselli, José Bento Sterman Ferraz, Luiz F. Brito

**Affiliations:** 1Department of Animal Sciences, Purdue University, West Lafayette, IN 47907, USA; 2Department of Veterinary Medicine, College of Animal Science and Food Engineering, University of São Paulo, Pirassununga, 13635-900 São Paulo, Brazil; miguel-has@hotmail.com (M.H.A.S.); fabianabressan@usp.br (F.F.B.); joapeler@usp.br (J.P.E.); jbferraz@usp.br (J.B.S.F.); 3Department of Biological Sciences, School of Sciences and Languages, São Paulo State University, Assis, 19806-900 São Paulo, Brazil; marcelo.fabio@unesp.br; 4Department of Applied Mathematics and Computer Science, Technical University of Denmark, 102500 Lyngby, Denmark; hajak@dtu.dk; 5College of Veterinary Medicine and Animal Science, University of Sao Paulo, 05508-270 São Paulo, Brazil; pietro.baruselli@gmail.com

**Keywords:** anti-Müllerian, biomarker, fertility rate, indicine, GWAS, heritability, ovarian reserve, reproduction, Zebu

## Abstract

**Simple Summary:**

Reproductive performance has direct economic implications on the beef cattle industry. The majority of Zebu breeds (e.g., Nellore; *Bos taurus indicus*) have poor reproductive performance compared to other cattle breeds. In this context, genetic progress can be achieved for fertility and reproduction through the use of genomic selection for indicator variables that efficiently assess the biological mechanisms underlying cattle fertility. Therefore, this study aimed to estimate genetic parameters (heritability and genetic correlations) and identify candidate genes associated with anti-Müllerian hormone levels (AMH) and antral follicle populations measured after estrous synchronization (AFP) in Nellore cattle. Our findings indicate that measuring circulating AMH in Nellore cattle enables the identification of animals with high genetic merit for superovulatory responses, as well as early selection of the best oocyte donors for in vitro embryo production. Both traits are heritable and influenced by a large number of important genes. Therefore, AMH and AFP can be used as indicator traits to genetically improve fertility rates in Nellore cattle and to identify better oocyte donors.

**Abstract:**

Reproductive efficiency plays a major role in the long-term sustainability of livestock industries and can be improved through genetic and genomic selection. This study aimed to estimate genetic parameters (heritability and genetic correlation) and identify genomic regions and candidate genes associated with anti-Müllerian hormone levels (AMH) and antral follicle populations measured after estrous synchronization (AFP) in Nellore cattle. The datasets included phenotypic records for 1099 and 289 Nellore females for AFP and AMH, respectively, high-density single nucleotide polymorphism (SNP) genotypes for 944 animals, and 4129 individuals in the pedigree. The heritability estimates for AMH and AFP were 0.28 ± 0.07 and 0.30 ± 0.09, and the traits were highly and positively genetically correlated (*r_G_* = 0.81 ± 0.02). These findings indicated that these traits can be improved through selective breeding, and substantial indirect genetic gains are expected by selecting for only one of the two traits. A total of 31 genomic regions were shown to be associated with AMH or AFP, and two genomic regions located on BTA1 (64.9–65.0 Mb and 109.1–109.2 Mb) overlapped between the traits. Various candidate genes were identified to be potentially linked to important biological processes such as ovulation, tissue remodeling, and the immune system. Our findings support the use of AMH and AFP as indicator traits to genetically improve fertility rates in Nellore cattle and identify better oocyte donors.

## 1. Introduction

Reproductive performance directly impacts the profitability of the beef cattle industry [1,2]. Precocity and fertility rates are lower in Zebu cattle (*Bos taurus indicus*) compared to Taurine breeds (*Bos taurus taurus*) [3]. The majority of beef cattle breeding programs focus on growth and carcass traits, which are usually easier to measure in comparison with female reproductive traits [4]. Furthermore, most reproductive traits have low to moderate heritability estimates, therefore, reduced genetic progress is usually attained per time unit [5,6]. Despite the high environmental influence on reproductive traits (i.e., low to moderate heritability estimates), there is enough genetic variability to enable genetic progress through direct selection [7,8]. Therefore, identifying novel phenotypes that better represent the biological mechanisms underlying reproductive performance could be of great value to beef cattle breeding programs, especially in Zebu cattle breeds raised in tropical conditions.

Anti-Müllerian hormone (AMH) regulates follicle selection [9], as well as antral follicle population (AFP) [10]. AMH is strongly associated with the ovarian response and embryo scores [11], and therefore in vitro fertilization [12]. In vitro fertilization and cattle embryo production are common practices in beef cattle production systems, especially in Zebu breeds [13,14,15]. The measurement of AMH concentration and its association with reproductive and fertility traits were widely studied in livestock, including cattle [16,17,18,19,20,21]. These studies indicated that circulating AMH levels are a useful indicator of fertility and reproductive performance. The role of AMH in different aspects of reproductive physiology is noteworthy, and it is involved with AFP [16,22], superovulation responses [23], in vitro embryo production [10,24,25], fertility traits (i.e., age at puberty and postpartum interval) [26], and longevity [27] in beef cattle. The definition and use of efficient biological indicators of fertility is a promising path to increase the success of reproductive technologies (e.g., embryo transfer, superovulatory responses), especially in Nellore cattle raised in tropical conditions.

Genome-wide association studies (GWAS) are frequently used in livestock studies aiming to detect causal mutations and candidate genes associated with traits of interest [28,29,30,31]. In this context, knowledge regarding the genomic regions, candidate genes, and metabolic pathways related to reproductive physiology and fertility indicators is paramount to the optimization of genomic selection to improve female reproductive efficiency [32,33]. In Holstein cattle, genomic regions associated with AMH levels were previously linked with fertility, superovulatory responses, and embryo development [20,34]. However, this is currently underexplored in Zebu cattle. The main objectives of this study were to estimate the genetic parameters for AMH and AFP using genomic information and to identify genomic regions, candidate genes, and metabolic pathways related to AMH and AFP traits in Nellore cattle using a high-density single nucleotide polymorphism (SNP) chip panel.

## 2. Methods

### 2.1. Ethical Statement

No approval from the local Ethics Committee was required for this study because the datasets were generated in previous experiments approved by the Bioethics Commission of the School of Veterinary Medicine and Animal Sciences of the University of Sao Paulo, Sao Paulo, Brazil [35,36].

### 2.2. Animals and Phenotypic Data

The datasets used were from three commercial beef cattle farms (Segredo, Engano, and CFM) located in the state of Mato Grosso do Sul in the midwestern region of Brazil. All animals were raised under similar environmental conditions and received mineral supplementation on a pasture-based production system, with ad libitum water. The pedigree dataset included 4129 animals (2601 females and 1528 males), in which 1032 were founders and 2883 individuals had both parents known. The pedigree spanned up to five generations according to suggestions that the inclusion of at least three generations is enough to predict accurate breeding values and estimate genetic parameters [37]. Furthermore, as genomic information was used, the relationships from more distant ancestors were also captured. All animals were from the same breeding program and were genetically related, as there was exchange of semen and breeding animals across the three farms. The average pedigree and genomic inbreeding coefficients were 0.008 and 0.010, respectively.

A total of 1099 females 425 (cows (16 months of age) and 674 heifers (14 months of age)) were measured for AFP. The total number of visible follicles (≥3 mm in diameter) in both ovaries were evaluated using ultrasonography equipment (7.5 MHz transrectal linear transducer, Mindray M5Vet; Mindray, Mahwah, NJ, USA). Animals were recorded at the beginning of the follicular wave (day 4 of the synchronization protocol after applying a progesterone intravaginal device, plus 2 mg of estradiol benzoate on day 0) [24]. The synchronization protocol was used to ensure that all the animals were measured at a similar reproductive stage, thereby reducing environmental influence on trait expression. Blood samples of 289 Nellore heifers were collected using vacuum tubes containing ethylenediamine tetraacetic acid—EDTA (Health Co, Canton, MA, USA) via jugular vein puncture to measure AMH concentrations. These heifers were, on average, 14.10 ± 0.03 months old, and all of them had AFP records. The AMH assay was conducted at the IgAc Laboratory (Institute Genese of Scientific Analyses, São Paulo, SP, Brazil) using the Bovine AMH enzyme-linked immunosorbent assay AL-114 kit (Ansh Labs, Webster, TX, USA), following the protocol described by Batista et al. [35].

### 2.3. Genotypic Quality Control

A total of 944 females (379 cows and 565 heifers) were genotyped using a high-density SNP chip panel (Illumina Inc., San Diego, CA, USA) containing 777,962 single nucleotide polymorphisms (SNP). Quality control was performed using the PREGSF90 program [38]. The following criteria were used for the exclusion of SNPs: minor allele frequency (MAF), SNP call rate, and animal call rate lower than 0.05, 0.90, and 0.90, respectively, extreme deviation (greater than 0.15) from the Hardy–Weinberg equilibrium, defined as the difference between the observed and expected frequency of heterozygotes [39], and markers located on the sex chromosomes or mitochondrial DNA. A total of 467,209 SNPs was found to be distributed across 29 autosomal chromosomes, and 917 samples remained for further analyses.

### 2.4. Statistical Analyses

#### 2.4.1. Variance Components, Genetic Parameters, and Breeding Value Prediction

A linear animal model and the Average Information Restricted Maximum Likelihood method (AI-REML) were used to estimate variance components, heritability, and genetic correlations using the AIREMLF90 package from the BLUPF90 software [40,41]. Genomic breeding values for both traits (AMH and AFP) were directly predicted using the single-step Genomic Best Linear Unbiased Predictor (ssGBLUP) procedure. A bivariate animal model was fitted as follows:**y** = **Xβ** + **Zα** + **e**,(1)
where **y** is the vector of individual observations for AMH and AFP, **β** is the vector associated with the fixed effect of contemporary group and the linear effect of age of the animal at the measurement, **α** is the vector of direct additive genetic effects, **X** and **Z** are the incidence matrices linking records to the **β** and **α** vectors, respectively, and **e** is the vector of the residual effects. The contemporary group was defined by concatenating the effects of the farm (3 levels), management group, birth year (2013–2017), and season of the year during which the measurement was taken. Additive genetic and residual effects were assumed to follow a normal distribution. The ssGBLUP is a modified version of the traditional BLUP, in which the inverse of the pedigree-based relationship matrix ($A^{-1}$) is replaced by the $H^{-1}$ matrix. The $H^{-1}$ was defined as follows [42,43]:(2)$$H^{-1}=A^{-1}+\left[\begin{array}{cc} 0 & 0\\ 0 & \tau G^{-1}-\omega A_{22}^{-1} \end{array}\right]$$
where $A^{-1}$ was previously defined, τ and ω are the scaling factors used to combine the genomic relationship matrix (G) and $A_{22}$, assumed as τ = 1.0 and ω = 0.7 (defined based on preliminary analyses) in order to reduce the bias of the estimates [44], $\omega A_{22}^{-1}$ is the inverse of the pedigree-based relationship matrix (A) for the genotyped animals, and $G^{-1}$ is the inverse of the G matrix, which was calculated as [45]:***G*** = **ZZ′/k**(3)
where **Z** is the matrix containing the centered genotypes accounting for the observed allelic frequencies and **k** is a scaling parameter, defined as 2∑p(1 − p), in which p is the observed allele frequency of each marker. The weighting factor can be derived either based on SNP frequencies [45], or by ensuring that the average diagonal of ***G*** is close to that of $A_{22}$ [46]. In order to minimize issues with matrix inversion, 0.05 of ***A*** was added to 0.95 of ***G***.

#### 2.4.2. Genome-Wide Association Analysis

GWAS was carried out for each trait based on the weighted ssGBLUP method (WssGBLUP) [29]. The same statistical models described to estimate the variance components and breeding values were used to identify the genomic windows associated with the traits, as described by Wang et al. [47], using the BLUPF90 program [40,41].

The POSTGSF90 program [38] was used to obtain SNP effects by back-solving the genomic estimated breeding values (GEBVs) for each trait. SNP effects and SNP weights were calculated following Wang et al. [47] based on three iterations. The GWAS results were reported as the proportion of the variance explained by a moving genomic window of 10 adjacent SNPs. Genomic windows that explained more than 1% of the total additive genetic variance were considered to be relevant, i.e., associated with AMH or AFP.

#### 2.4.3. Functional Analyses

The candidate gene list from the genomic regions that explained at least 1% of genetic variance was annotated considering upstream and downstream intervals of 100 kb (threshold defined based on the level of linkage disequilibrium in the population) via the *BioMart* tool using the Ensembl Genes and the *Bos taurus taurus* ARS-UCD1.2 reference genome [48]. The DAVID v6.8 [49] software was used to perform the enrichment analysis according to the similarity of the biological processes and Kyoto Encyclopedia of Genes and Genomes (KEGG) pathways in which they are involved in (*p* ≤ 0.05; False Discover Rate ≤ 10), using all candidate genes identified for AMH and AFP. Furthermore, important SNPs (from the key genomic windows) were further explored using the Animal QTL Database (AnimalQTLdb) [50].

## 3. Results

### 3.1. Variance Component and Genetic Parameter Estimates

The descriptive statistics and genetic parameter estimates for AMH and AFP are presented in Table 1. The estimate of the genetic correlation between AMH and AFP was positive and of high magnitude (*r_G_* = 0.81 ± 0.02).

### 3.2. GWAS and Functional Analyses

A total of 13 genomic regions located on BTA1, BTA3, BTA5, BTA7, BTA8, BTA10, BTA11, BTA18, BTA22, and BTA25 were identified for AMH (Table 2). Additionally, 18 genomic windows located on BTA1, BTA2, BTA4, BTA6, BTA8, BTA11, BTA14, BTA21, BTA26, BTA28, and BTA29 accounted for at least 1% of the total genetic variance for AFP (Table 3).

The chromosomes BTA5 and BTA7 demonstrated the most important genomic regions associated with AMH. For AFP, the two highest peaks were identified on BTA1 (at 109.1 Mb) and BTA26 (at 45.7 Mb), which accounted for 6.24% and 7.33% of the total additive genetic variance, respectively. The highest peak for AMH was found on BTA5 (at 97.2 Mb) explaining 5.22% of the total additive genetic variance. Two overlapping regions located on BTA1 (64.9–65.0 Mb and 109.1–109.2 Mb) were found for the two traits, as illustrated in Figure 1 and Figure 2.

Totals of 26 and 22 positional candidate genes were identified for AMH and AFP, respectively. For AMH, one (*PLXNC1*) and four (*GRP19*, *CREBL2*, *DUSP16*, and *BORCS5*) positional genes were detected on BTA5, located around 24.0 Mb and between 97.1 and 97.4 Mb, respectively. A group of genes (*ADAM12*, *TIAL1*, *BAG3*, and *INPP5F*) were found to be harbored within the genomic region located on BTA26 for AFP. The overlapping regions described above shared three candidate genes, namely, *RSRC1*, *GPR156*, and *LRRC58*. Five biological processes and one pathway were enriched (*p* ≤ 0.05), including pathways associated with the ovulation process (Table 4).

## 4. Discussion

The circulating concentrations of AMH are highly variable among mammalian species. In the current study, the AMH mean agreed with Zebu cattle reports, which ranged from 1.20 to 1.60 ng/mL [21,24,48]. These values were higher when compared with those reported for Taurine breeds (0.78 ng/mL [52]). In this study, high AMH plasma concentration was associated with greater AFP (>25). In this context, Morotti et al. [52] reported a strong (*r_G_* = 0.88) genetic correlation between AMH and AFP, which was in agreement with our current findings, highlighting the importance of measuring AMH concentration to identify precocious cows through follicular and ovulatory responses [53]. AFP was quantified after an estrous synchronization protocol, thereby indicating the response to a drug-induced treatment. This was done because estrous synchronization is a common practice in the cattle industry and also facilitates the measurement of individual animals at the same stage of reproduction, thus reducing environmental influence that could be difficult to account for in the statistical models.

Recently, Nawaz et al. [34] obtained higher heritability estimates for AMH based on pedigree (0.43 ± 0.07) or genomic (0.36 ± 0.03) information in Holstein cattle compared to the estimates found in the present study. Gobikrushanth et al. [20] also reported a high (0.46 ± 0.31) heritability estimate for AMH in Holstein cows. The high standard error in the last study might be due to the low (n = 198) number of cows with phenotypic records, data structures, and transformation of phenotypic data. The AMH heritability estimate observed in Nellore heifers (*h*^2^ = 0.28 ± 0.07) was lower when compared to the literature reports, however, a lower standard error was also observed in the current study. This moderate heritability indicated that substantial genetic progress could be achieved through direct genetic selection.

Júnior et al. [54] reported higher heritability estimates for AFP (0.49 ± 0.09) in Nellore cattle compared to our findings. Moderate to high positive genetic correlations between AMH and AFP were reported to range from 0.56 to 0.68 in Nellore cattle and 0.73 to 0.90 in Holstein animals [52,55]. Since AMH is already certified as an endocrine marker of ovarian reserve in women [56], this significant positive correlation indicated that both traits could be incorporated into Nellore cattle breeding programs. Some authors [12,23,53] indicated a practical advantage in the use of AMH instead of predicting AFP using an ultrasound approach. The levels of circulating AMH are stable during the estrous cycle [53], therefore, it can be measured at any time during that period. AMH is expected to be a useful trait to identify animals that are better oocyte donors in superovulation protocols. Nawaz et al. [34] reported significant genomic regions on BTA11 (92.8 to 97.1 Mb) and BTA20 (25.0 to 26.3 Mb). In this present study, genomic regions on BTA11 (around 6 Mb) for AMH and BTA11 (around 48 Mb) for AFP were observed. Moreover, these authors [34] reported gene members of the *TGF-β* family involved in follicular development, cell proliferation, steroidogenesis, and ovulation.

Similar to our findings, three genomic windows located in BTA8 (4.16–7.66 Mb), BTA11 (69.4–73.6 Mb), and BTA22 (11.9–15.9 Mb) were associated with AFP in Nellore heifers [54]. These regions harbor important candidate genes (*ADAM29*, *CTSB*, *SOX7*, *PPP1CB*, *CLIP4*, and *CCK*) associated with fertility traits. The significant peak on BTA26 around 46 Mb harbors the gene *ADAM12* (Adam metallopeptidase domain 12), which is associated with several biological activities, such as regulating remodeling of extracellular matrix, modulation of cell morphological changes, satellite cell activation, regulation of myogenesis and adipogenesis in beef cattle [57], and regulation of *TGF-β1* [58], a gene involved in follicular development, cell proliferation, steroidogenesis, and ovulation [59]. Evidence exists regarding the function of the metalloprotease domain (ADAM) gene family in the ovarian follicle, follicular development, and ovarian organization [60,61]. The three proteases (*ADAM9*, *ADAM10*, and *ADAM12*) together seem to regulate the breakdown of matrix and differentiation of granulosa cells prior to ovulation [62]. The *BAG3* (BTA26:39.7–39.8 Mb) and *TIAL1* (BTA26:39.6–39.7 Mb) genes were also annotated. *TIAL1* acts on post-transcriptional regulation encoding protein members of the RNA binding family, and has also been associated to the development of primordial germ cells of gametes (i.e., sperm and oocytes) in rodents [63]. In addition, *BAG3* was annotated in proximity to a QTL affecting daughter pregnancy rate in Nellore beef cattle [3].

The enrichment analysis (Table 4) highlighted five biological processes and one KEGG pathway related to AMH and AFP. The response to an external stimulus term comprises any mechanism that regulates or modifies the gene expression or secretion or enzyme production as a result of an external stimulus (e.g., nutritional, environmental, maternal behavior) or epigenomics [64]. In livestock, various studies investigated the molecular epigenetic mechanisms that regulate the expression of certain genes potentially involved in reproductive traits’ expression, sometimes as a response to an external stimulus [65,66,67,68]. It is important to highlight that multiple external factors can leave epigenetic marks that could affect further generations and may be related to the activity and levels of AMH and AFP. The group of genes involved in this biological process (response to external stimulus; *IL1R2*, *SERINC5*, *PLXNC1*, *THBS4*, and *CBS*) are implicated in the ovulation process, tissue remodeling, immune system, and embryo preimplantation [69,70]. *PLXNC1* (Plexin C1) is associated with the immune system [50] and belongs to a subfamily of plexin genes which function as receptors for semaphorins to influence neuronal development and function. The immune system regulates physiological interactions to support internal protection, health, and survival of the embryo. Herein, the interaction between the immune and reproductive systems are of particular interest [71]. The main interaction between the systems is related to endocrine and immune response during gestation [72]. Particularly, *IL1R2* is a glycoprotein expressed on monocytes, neutrophils, and T and B lymphocytes. Part of the interleukin family, the type 2 form of *IL-1,* was extensively related to ovulation during follicular growth [69]. *THBS4* is a calcium-binding protein that modulates cellular phenotype during tissue genesis and remodeling [73]. In this regard, ovulation was suggested to be a tissue remodeling process, annotated as a significant biological process term (*p* = 0.02). The genes *CNTFR* and *ATP1A1* are associated with productive efficiency (e.g., average daily gain, feed efficiency) [74] and heat tolerance [75,76,77] indicators. Selection for fertility along with production traits is, therefore, readily justifiable. *ATP1A1* was reported to be related to heat tolerance [71], in which heat stress response can affect cattle reproductive performance [76,77]. *ATP1A1* may also influence mastitis resistance [78].

The KEGG protein processing in the Mitogen-Activated Protein Kinase (MAPK) signaling pathway is involved in a conserved module that controls various cellular events and biological processes, such as embryogenesis, cell differentiation, cell proliferation, cell death, short-term changes required for homeostasis, and acute hormonal responses [79,80,81]. Also, this pathway is associated with preimplantation embryogenesis [82], defined as the time interval from conception to nidation or attachment of the embryo to the uterus, a crucial stage in successful pregnancies related to embryonic health, and therefore reproductive efficiency [83].

There are known QTLs located in the genomic regions identified in this study, including associations with calving performance (e.g., calving ease, calf size) [84]. In agreement with our findings, previous studies in Angus and Hereford cattle detected the MAPK signaling pathway as having a pleiotropic effect on birth weight, calving ease (direct and maternal), and calving performance traits [85,86,87,88]. Likewise, QTLs linked to gestation length, the interval to first estrus after calving [89], conception rate [90], and heifer pregnancy [91] overlapped with various genomic regions identified for AMH and AFP. The anti-Mullerian hormone receptor type 2 (*AMHR2*) gene was not identified as significant in this study, but was reported as an important gene related to AMH [9,92,93]. Surprisingly, a genomic window harboring the AMHR2 gene, located at BTA5:26,591,483–26,597,710, only explained 0.001% of the total additive genetic variance for AMH. However, a genomic region identified on BTA5 may capture the effect of this gene through linkage disequilibrium (Figure 1), or prove that the polymorphisms in other genes involved in the AMH metabolic process are of greater importance. In this context, Pierucci et al. [94] investigated the association between polymorphisms in the *AMH* gene and early pregnancy occurrence and age at first calving in Nellore cattle, but also did not find significant additive effects of the three SNPs investigated.

Despite the fact that AMH and AFP are highly genetic correlated traits (0.81 ± 0.02), a limited number of overlapping genomic regions were identified, indicating that the genes with pleiotropic effects for AMH and AFP might be genes with small effects. Furthermore, based on the heritability estimates and large number of genomic regions with small effects, AMH and AFP seem to be traits of complex inheritance regulated by a large number of small-effect genes.

Genetic selection for AMH and AFP in Nellore cattle is expected to improve reproductive performance, especially superovulatory responses. Furthermore, breeding values for these traits can be used to select the best oocyte donors for in vitro embryo production. The number of genotyped animals with phenotypes for AMH is a limiting factor that needs to be increased in future studies, as this impacts the identification of important genomic regions with smaller effects. However, the use of a high-density SNP panel may have minimized this issue, as the average linkage disequilibrium between markers was greater compared to a lower density SNP panel. Further studies using datasets from independent populations, as well as larger datasets in this current population, should be performed to validate the findings of this study. AFP can be measured at a reasonably low cost per individual and, as AMH and AFP are highly and positively genetic correlated, AFP could be a feasible indicator for inclusion in breeding schemes.

## 5. Conclusions

Our findings indicated that anti-Müllerian hormone levels and antral follicle populations (measured after a synchronization protocol) are heritable traits in Nellore cattle and can be improved through genetic and genomic selection. Furthermore, breeding values for both traits can be used to identify females that best respond to superovulation that are better oocyte donors for in vitro embryo production. Substantial indirect genetic gains are expected by selecting for only one of the two traits, as a high positive correlation was observed between them. A total of 31 genomic regions were identified to be associated with AMH or AFP, and two genomic regions located on BTA1 overlapped between the traits. Various candidate genes were identified to be potentially linked to important biological processes such as ovulation, tissue remodeling, and the immune system. Anti-Müllerian hormone levels and antral follicle populations could be used as indicator traits to genetically improve fertility rates in Nellore cattle and to identify better oocyte donors. Future studies should also investigate the genomic backgrounds of these traits in the absence of an estrous synchronization protocol.

## Figures and Tables

**Figure 1 animals-10-01185-f001:**
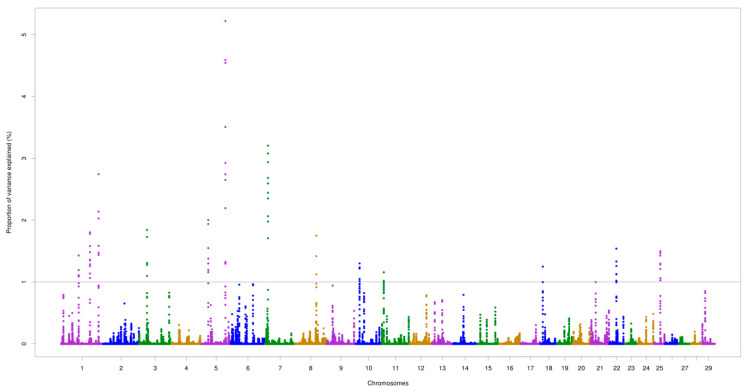
Genomic regions for anti-Mullerian hormone (AMH) levels in Nellore cattle. The grey line represents the threshold (1%) of the proportion of total additive genetic variance accounted for by each genomic window.

**Figure 2 animals-10-01185-f002:**
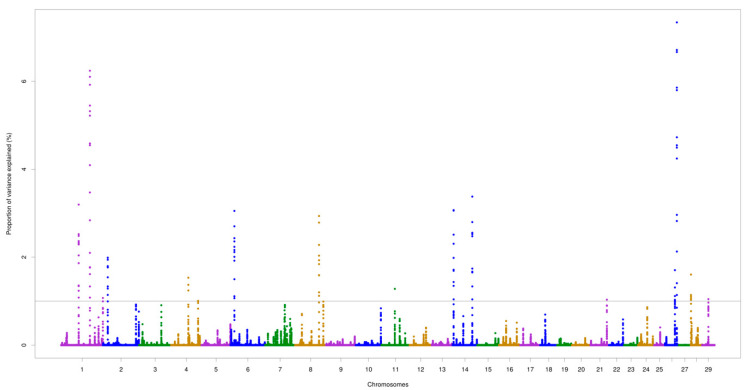
Genomic regions for antral follicle populations (AFP) in Nellore cattle. The grey line represents the threshold (1%) of the proportion of total additive genetic variance accounted for by each genomic window.

**Table 1 animals-10-01185-t001:** Descriptive statistics, additive genetic variance (*σ*^2^*_a_*), residual variance (*σ*^2^*_e_*), heritability (*h*^2^), and standard error (SE) estimated for anti-Müllerian hormone (AMH) and antral follicle populations measure after estrous synchronization (AFP) in Nellore cattle.

Trait	Mean	SD ^3^	*σ*^2^*_a_* ± SE	*σ*^2^*_e_* ± SE	*h*^2^ ± SE
AMH ^1^	1.10	0.36	0.16 ± 0.10	0.40 ± 0.10	0.28 ± 0.07
AFP ^2^	12.49	4.09	3.02 ± 2.02	7.13 ± 1.96	0.30 ± 0.09

^1^ Concentration in ng/mL; ^2^ number of antral follicles (n); ^3^ standard deviation.

**Table 2 animals-10-01185-t002:** Candidate genes located in the genomic windows that accounted for more than 1% of the total additive genetic variance for anti-Müllerian hormone levels in Nellore cattle.

Chr:Position (bp) ^1^	Candidate Genes	Var (%) ^2^
5:97,149,440–97,360,900	*GPR19, CREBL2, DUSP16, BORCS5*	5.22
7:13,239,159–13,465,265	*-*	3.20
1:143,011,858–143,226,874	*PDE9A, WDR4, NDUFV3, PKNOX1, CBS*	2.74
5:23,779,410–24,021,506	*PLXNC1*	2.00
3:26,810,702–27,023,916	*ATP1A1*	1.84
1:109,052,581–109,300,000	*RSRC1*	1.80
8:75,771,507–75,987,944	*FAM219A, DANI1, ENHO, CNTFR, RPP25L, DCTN3*	1.75
22:26,343,690–26,562,220	*-*	1.54
25:21,973,041–22,186,659	*CACNG3*	1.49
1:64,865,063–65,123,147	*GPR156, LRRC58*	1.43
10:10,957,392–11,173,990	*MTX3, THBS4, SERINC5*	1.30
18:7,065,051–7,298,879	*-*	1.25
11:6,614,691–6,831,396	*MAP4K4, IL1R2*	1.15

^1^ Chromosome; ^2^ Var (%) = proportion of the total additive genetic variance.

**Table 3 animals-10-01185-t003:** Candidate genes located in the genomic windows that accounted for more than 1% of the total additive genetic variance for antral follicle populations in Nellore cattle.

Chr:Position (bp) ^1^	Candidate Genes	Var (%) ^2^
26:45,605,887–45,832,830	*ADAM12*	7.33
1:109,036,868–109,295,720	*RSRC1*	6.24
14:68,529,242–68,764,179	-	3.37
1:64,905,320–65,192,584	*GPR156, LRRC58, FSTL1*	3.19
14:4,866,037–5,110,734	-	3.07
6:13,887,045–14,107,585	-	3.05
8:86,936,023–87,174,520	*SYK*	2.93
2:16,843,125–17,099,561	*ZNF385B*	1.98
2:16,761,007–16,999,128	*CWC22*	1.79
26:39,682,007–39,898,965	*TIAL1, BAG3, INPP5F*	1.70
28:478,980–713,335	*RHOU*	1.60
4:77,214,698–77,441,075	*POLM, BLVRA, COA1*	1.53
6:13,909,732–14,132,804	-	1.49
11:48,459,747–48,702,594	*REEP1, MRPL35, IMMT*	1.28
1:157,802,376–158,023,994	*GPX5*	1.07
29:23,486,872–23,711,141	-	1.04
21:62,231,714–62,455,313	-	1.03
4:10,571,640–10,785,338	*HEPACAM2, VPS50, CALCR*	1.00

^1^ Chromosome; ^2^ Var (%) = proportion of the total additive genetic variance.

**Table 4 animals-10-01185-t004:** Most significant (*p* ≤ 0.05) biological processes (BP) and KEGG pathways for anti-Müllerian hormone levels and antral follicle populations in Nellore cattle.

Type ^1^	Term	Candidate Genes	*p*-Value	FDR
BP	GO:0048771~tissue remodeling	*THBS4, CBS, SYK*	0.024	3.2
BP	GO:0051301~cell division	*TIAL1, HEPACAM2, DCTN3, THBS4*	0.036	4.4
BP	GO:0007005~mitochondrion organization	*MTX3, IMMT, COA1, RHOU, MRPL35*	0.040	4.8
BP	GO:0009605~response to external stimulus	*IL1R2, SERINC5, PLXNC1, BAG3, FSTL1, THBS4, CBS, SYK*	0.045	5.2
BP	GO:0051049~regulation of transport	*IL1R2, ATP1A1, CACNG3, REEP1, RHOU, CREBL2, SYK*	0.048	5.4
KEGG	bta04010:MAPK signaling pathway	*MAP4K4, DUSP16, CACNG3*	0.049	3.8

^1^ BP: Biological process; KEGG: Kyoto Encyclopedia of Genes and Genomes (KEGG) pathway; FDR: False Discovery Rate based on the Benjamini–Hochberg (BH) method [51].

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
