# Peer review of "Genetic Parameters and Genome-Wide Association Studies for Anti-Müllerian Hormone Levels and Antral Follicle Populations Measured After Estrus Synchronization in Nellore Cattle"

_animals, 2020, doi:10.3390/ani10071185_

Round 1

Reviewer 1 Report

Dear authors,

The manuscript is well written and discussed.

Although the number of animals genotyped is not ideal, we understand the difficulty to collect reproductive phenotypes, specially promising ones as are the ones herein reported. The manuscript has credit to be published. Some comments were done in order to better discuss the results obtained.

  1. Order the genomic windows (Tables 2 and 3) by percentage of additive genetic variance explained, not by chromosome position. We need to easily identify the most important regions influencing the traits.
  2. Although the traits studied are highly genetic correlated, the major genes influencing them are not. There are just two overlapping windows. It means that the genes with pleiotropic effects that influence both traits are genes with minor effect. It is important to report since it better describes the genetic architecture of the traits. Moreover, it gives useful guides to construct low-density SNP chips to be used in genomic selection, in terms of which trait is better to be selected. A significant SNP for one trait might not have the same effect in the other trait, although they are correlated.
  3. The authors used just another GWAS article [53] in the Discussion. There are other two that should be used [19 and 34] since they worked with the same trait in other cattle breeds. Are the regions the same? Are they different?
  4. The fact that the AHM and AHMR2 genes were not signalized is not a real concern. However, it must be better discussed. The authors suggest a possible linkage with the window in BTA5. It is an explanation. However, it is important to see if AHM and AHMR2 genes have circulating SNPs from the chip first. If there were SNPs in the genes and they were not significant, it just means that polymorphism in other genes of the metabolic process are more important.

The linkage discussion should be maintained. It is just another way to explain the results. Moreover, there is a recent article “Amh Polymorphisms and their association with traits indicative of sexual precocity in Nelore heifers” by Pierucci et al (2019) that reports lack of association of AHM SNPs and reproductive traits in Nellore cattle. It should be useful in Discussion.

  1. A major concern is about the metabolic pathways found. The pathways are important to describe the metabolism underling the trait (and it is good and important). However, so much attention was given to the discuss of the biological processes. They are just statistically significant genes group in a pathway. It does not make them more important than a gene explaining high additive variance. My special concern is that they were inserted in conclusion as main findings. It is not true. There are other genes with greater importance such as ADAM12 for AFP that would contribute more in a selection process. It just need to be better addressed and the text must be adapted.

Author Response

Dear Reviewer,

We would first like to thank you for your thorough review of this manuscript and thoughtful suggestions, which have been of great benefit. The manuscript was revised substantially to address the main concerns of the reviewers. A point-by-point response to the reviewers’ comments and suggestions is provided.

Reviewer : The manuscript is well written and discussed. Although the number of animals genotyped is not ideal, we understand the difficulty to collect reproductive phenotypes, specially promising ones as are the ones herein reported. The manuscript has credit to be published. Some comments were done in order to better discuss the results obtained.

AU: Thank you for all the great comments and suggestions. We also agree with your comment with regards to the sample size and we are working towards increasing it for future studies.

Reviewer : Order the genomic windows (Tables 2 and 3) by percentage of additive genetic variance explained, not by chromosome position. We need to easily identify the most important regions influencing the traits.

AU: This was done as suggested. Please see Tables 2 and 3.

Reviewer : Although the traits studied are highly genetic correlated, the major genes influencing them are not. There are just two overlapping windows. It means that the genes with pleiotropic effects that influence both traits are genes with minor effect. It is important to report since it better describes the genetic architecture of the traits. Moreover, it gives useful guides to construct low-density SNP chips to be used in genomic selection, in terms of which trait is better to be selected. A significant SNP for one trait might not have the same effect in the other trait, although they are correlated.

AU: Thank you for the great point. We have included some sentences on this regard in the Discussion section (all the changes are highlighted).

Reviewer : The authors used just another GWAS article [53] in the Discussion. There are other two that should be used [19 and 34] since they worked with the same trait in other cattle breeds. Are the regions the same? Are they different?

AU: Thank you. The references [19] and [53] are related to the traits studied here, but they did not perform GWAS. The reference [34] has been properly discussed as suggested. Please see L233-236. 

Reviewer : The fact that the AHM and AHMR2 genes were not signalized is not a real concern. However, it must be better discussed. The authors suggest a possible linkage with the window in BTA5. It is an explanation. However, it is important to see if AHM and AHMR2 genes have circulating SNPs from the chip first. If there were SNPs in the genes and they were not significant, it just means that polymorphism in other genes of the metabolic process are more important.

AU: Thank you for the comment. This has been discussed in the revised manuscript (all changes are highlighted).

Reviewer : The linkage discussion should be maintained. It is just another way to explain the results. Moreover, there is a recent article “Amh Polymorphisms and their association with traits indicative of sexual precocity in Nelore heifers” by Pierucci et al (2019) that reports lack of association of AHM SNPs and reproductive traits in Nellore cattle. It should be useful in Discussion.

AU: Thank you so much. This paper has been included in our discussion. 

Reviewer : A major concern is about the metabolic pathways found. The pathways are important to describe the metabolism underling the trait (and it is good and important). However, so much attention was given to the discuss of the biological processes. They are just statistically significant genes group in a pathway. It does not make them more important than a gene explaining high additive variance. My special concern is that they were inserted in conclusion as main findings. It is not true. There are other genes with greater importance such as ADAM12 for AFP that would contribute more in a selection process. It just need to be better addressed and the text must be adapted.

AU: Thank you for pointing it out. The manuscript has been modified according to your suggestion.

Reviewer 2 Report

The authors aimed to estimate genetic parameters (heritability and genetic correlation) and identify candidate genes associated with anti-Müllerian hormone level (AMH) and antral follicle population (AFP) in Nellore cattle, and acquired some interesting results. I think it is valuable for cattle breeding. But, in the manuscript, there are some minor errors that need to be revised before publication.

  1. Page 2 (L60-L61), “as well as antral follicle population (AFP)” , pleaseprovide references to support the sentence.
  2. Page 2 (L62), “in-vitro” , in this article, the word is either italicized or not, please unify them.
  3. For results, you can use subheadings to better illustrate the results of each section.
  4. Page 9 (L234),“The response to external stimulus term ……or epigenomics”, please provide the entire reference.
  5. Page 9 (L249-L251),“The THBS4 is a calcium-binding protein ……as a significant biological process term” Please, provide the entire reference.
  6. Page 9 (L265),”Various of these QTLs were associated”, it is an error.

Author Response

Dear Reviewer,

We would first like to thank you for your thorough review of this manuscript and thoughtful suggestions, which have been of great benefit. The manuscript was revised substantially to address the main concerns of the reviewers. A point-by-point response to the reviewers’ comments and suggestions is provided.

Reviewer : The authors aimed to estimate genetic parameters (heritability and genetic correlation) and identify candidate genes associated with anti-Müllerian hormone level (AMH) and antral follicle population (AFP) in Nellore cattle, and acquired some interesting results. I think it is valuable for cattle breeding. But, in the manuscript, there are some minor errors that need to be revised before publication.

AU: Thank you for all the valuable suggestions and comments, which greatly contributed to improve the quality of our manuscript.

Reviewer : Page 2 (L60-L61), “as well as antral follicle population (AFP)”, please provide references to support the sentence.

AU: This was done as suggested. Please see L61.

Reviewer : Page 2 (L62), “in-vitro” , in this article, the word is either italicized or not, please unify them.

AU: This was done as suggested.

Reviewer : For results, you can use subheadings to better illustrate the results of each section.

AU: This was done as suggested.

Reviewer : Page 9 (L234),“The response to external stimulus term ……or epigenomics”, please provide the entire reference.

AU: This was done as suggested. Please see L257.

Reviewer : Page 9 (L249-L251),“The THBS4 is a calcium-binding protein ……as a significant biological process term” Please, provide the entire reference.

AU: This was done as suggested. Please see L273.

Reviewer : Page 9 (L265) ”Various of these QTLs were associated”, it is an error.

AU: These sentences were rephrased. Please see L287-288.

Reviewer 3 Report

Comments on the manuscript Genetic parameters and genome-wide association studies for anti-Müllerian hormone level and antral follicle population in Nellore  cattle

This study deals with the genetic parameters of two reproductive-associated traits in a zebu population.

The authors used

The main issue from my point of view is that individuals were synchronized. Therefore, the authors are not evaluating AFP per se. Moreover, they are evaluating the individual reaction to a drug-induced treatment. Therefore, the conclusions of all the study (in AFP) should be reflected as-is.

Additionally, several methodological issues make this study experimentally weak, from my point of view.

First, the pedigree used. The authors stated that pedigree depth is three generations. A more clear characterization is necessary including estimation of ECG, MAX gen, and pedigree completeness indexes. There is also no mention of the relationship between individuals or farms. Are they belonging to the same “genetic base” or they can be analyzed as clustered populations??

Second, the model employed is extremely simple. Why the authors do not take into account the season of the year, year, etc. I'm pretty sure that such simplification of the model will bias the results. Even more when the error is near to 40 and 60% of the variance in each trait (approximately).

This comment is also valid for the GWAS model. But I don’t understand why the authors don’t test the p-value of each candidate gene individually to get more reliable conclusions.   

Another important fact is that authors discarded ECAX. Why is that?? There are several reports (such as 10.2527/jas.2012-6807) mentioning ECAX genes involved in fertility traits in cattle.

During Gwas, which correction was made?. There is no p-value of the GWAS nor threshold. Only the genomic intervals explaining 1% of the variance (based on a weak model) were taken. I believe that that methodology is not strong enough,

The functional analysis methodology is not clear. Which genes were included in DAVID?. Only those observed in the overlapping area between the two GWAS??. How many genes were included in total Information is missing. The results expressed in p-values were corrected by any false discovery rate ?.

Additionally, the functional association with genes is weak from my point of view. For example, the MAPK pathway showed a 0.05 p-value. I suppose that it was higher to 0.05 (since only two digits were used ), and therefore, It is NS, and cannot be used to support part of the discussion.

Finally, the fact that not there was no association with AMHR´s suggests that the variability among individuals in such receptors is low. That sounds weird to me (But it is an opinion)

To conclude, I believe that experimental design should be substantially improved before any further consideration to be published in ANIMALS.

Author Response

Dear Reviewer,

We would first like to thank you for your thorough review of this manuscript and thoughtful suggestions, which have been of great benefit. The manuscript was revised substantially to address the main concerns of the reviewers. A point-by-point response to the reviewers’ comments and suggestions is provided.

Reviewer : The main issue from my point of view is that individuals were synchronized. Therefore, the authors are not evaluating AFP per se. Moreover, they are evaluating the individual reaction to a drug-induced treatment. Therefore, the conclusions of all the study (in AFP) should be reflected as-is.

AU: We agree with the reviewer and have clarified it in the text. However, it is worth pointing out that this is a common practice in beef cattle reproductive management and the required reproductive performance would be expressed under similar conditions. Without synchronization, it would be very challenging to measure individual animals at the same state and eliminate all the environmental factors influencing AFP expression. Please see L106-108, L211-215, and L318.

Reviewer : Additionally, several methodological issues make this study experimentally weak, from my point of view.

AU: Thank you for the direct feedback. We have implemented commonly-used methods in all the analysis, as clarified in the revised manuscript and in the comments below. We agree that our sample size is small, but as also indicated by Reviewer #1, these are difficult-to-measure traits and of importance to the field. We have substantially revised the manuscript and included some sentences about the limitations of the study. All changes are highlighted in the manuscript.

Reviewer : First, the pedigree used. The authors stated that pedigree depth is three generations. A clearer characterization is necessary including estimation of ECG, MAX gen, and pedigree completeness indexes. There is also no mention of the relationship between individuals or farms. Are they belonging to the same “genetic base” or they can be analyzed as clustered populations?

AU: This is an excellent suggestion and we have better described the pedigree and the animal resources in the revised manuscript. Please see L93-100.

Reviewer : Second, the model employed is extremely simple. Why the authors do not take into account the season of the year, year, etc. I'm pretty sure that such simplification of the model will bias the results. Even more when the error is near to 40 and 60% of the variance in each trait (approximately).

AU: All these variables mentioned were taken into account by the Contemporary Group effect as there was a statistically significant interaction among those effects. This model has been developed based on a backwards selection approach and all the significant effects were included in the model presented in the manuscript. This has been clarified in the revised manuscript. Please see L135-137.

Reviewer : This comment is also valid for the GWAS model. But I don’t understand why the authors don’t test the p-value of each candidate gene individually to get more reliable conclusions.  

AU: I assume the reviewer is referring to p-values of individual SNPs, but we will address both SNPs and genes. First, we used the single-step GWAS method, which tests all SNPs simultaneously and therefore, there isn’t an exact p-value as we would obtain in a single SNP regression analysis. The BLUPF90 software developers have recently released a software version that calculates an approximate p-value (Aguilar et al., 2019a,2019b). However, the results were not stable, especially when using different values of tuning parameters in the construction of the H matrix (tau and omega). Similar observation was independently made by our colleagues in Canada and China using dairy and beef cattle datasets, respectively. As this option in BLUPF90 has been recently released and we are still not confident on the parameters that might affect the p-value calculation in a ssGWAS setting, we prefer to keep “% of variance explained”, which is a metric well accepted and used in the literature (e.g., Wang et al., 2012; 2014; Irano et al., 2016; Lemos et al., 2016; Oliveira Silva et al., 2017; Marques et al., 2018; Silva et al., 2019; Grigoletto et al., 2019; 2020). We will definitely reconsider it in future studies. Secondly, the p-values for the gene tests in the pathway analysis have been presented in the Tables.  

References:

Aguilar, I., Legarra, A., Cardoso, F., Masuda, Y., Lourenco, D., & Misztal, I. (2019a). Exact p-values for large-scale single step genome-wide association, with an application for birth weight in American Angus. bioRxiv, 555243. doi: https://doi.org/10.1101/555243

Aguilar, I., Legarra, A., Cardoso, F., Masuda, Y., Lourenco, D., & Misztal, I. (2019b). Frequentist p-values for large-scale-single step genome-wide association, with an application to birth weight in American Angus cattle. Genetics Selection Evolution, 51(1), 1-8. doi: https://doi.org/10.1186/s12711-019-0469-3

Wang, H., Misztal, I., Aguilar, I., Legarra, A., & Muir, W. M. (2012). Genome-wide association mapping including phenotypes from relatives without genotypes. Genetics Research, 94(2), 73-83.

Wang, H., Misztal, I., Aguilar, I., Legarra, A., Fernando, R. L., Vitezica, Z., ... & Muir, W. M. (2014). Genome-wide association mapping including phenotypes from relatives without genotypes in a single-step (ssGWAS) for 6-week body weight in broiler chickens. Frontiers in Genetics, 5, 134.

Wang, H., Misztal, I., Aguilar, I., Legarra, A., Fernando, R. L., Vitezica, Z., ... & Muir, W. M. (2014). Genome-wide association mapping including phenotypes from relatives without genotypes in a single-step (ssGWAS) for 6-week body weight in broiler chickens. Frontiers in Genetics, 5, 134.

Lemos, M. V., Chiaia, H. L. J., Berton, M. P., Feitosa, F. L., Aboujaoud, C., Camargo, G. M., ... & Mazalli, M. R. (2016). Genome-wide association between single nucleotide polymorphisms with beef fatty acid profile in Nellore cattle using the single step procedure. BMC genomics, 17(1), 213.

Medeiros de Oliveira Silva, R., Bonvino Stafuzza, N., de Oliveira Fragomeni, B., Miguel Ferreira de Camargo, G., Matos Ceacero, T., Noely dos Santos Gonçalves Cyrillo, J., ... & Misztal, I. (2017). Genome-wide association study for carcass traits in an experimental Nelore cattle population. PLoS One, 12(1), e0169860.

Marques, D. B., Bastiaansen, J. W., Broekhuijse, M. L., Lopes, M. S., Knol, E. F., Harlizius, B., ... & Lopes, P. S. (2018). Weighted single-step GWAS and gene network analysis reveal new candidate genes for semen traits in pigs. Genetics Selection Evolution, 50(1), 40.

Grigoletto, L., Brito, L. F., Mattos, E. C., Eler, J. P., Bussiman, F. O., Silva, B. D. C. A., ... & Ferraz, J. B. S. (2019). Genome-wide associations and detection of candidate genes for direct and maternal genetic effects influencing growth traits in the Montana Tropical® Composite population. Livestock Science, 229, 64-76.

Grigoletto, L., Ferraz, J., Oliveira, H. R., Eler, J. P., Bussiman, F. O., Abreu Silva, B. C., ... & Brito, L. F. (2020). Genetic Architecture of Carcass and Meat Quality Traits in Montana Tropical® Composite Beef Cattle. Frontiers in genetics, 11, 123. 

Reviewer : Another important fact is that authors discarded ECAX. Why is that?? There are several reports (such as 10.2527/jas.2012-6807) mentioning ECAX genes involved in fertility traits in cattle.

AU: We are not aware of the annotation of ECAX genes in the cattle genome and therefore, we assumed the reviewer might be referring to BTAX genes. We agree with the reviewer’s suggestion, however, there are some reasons for removing the non-autosomal chromosomes from the analyses in our specific case. First, there is a reduced SNP coverage in BTAX based on the SNP chip used as well as differences in the minor allele frequency of SNPs located on the BTAX. The SNPs located in BTAX also require special handling in the GWAS analyses, especially when the effect of all SNPs are estimated simultaneously, which is the case of ssGWAS. Considering the reduced dataset and that it included both males and female genotypes, including the BTAX could increase the rate of false positives as there might be problems with genotype calling for hemizygous males as a result of the lower intensity of some BTAX variants. In brief, we agree with the reviewer and we will include the BTAX chromosome when performing GWAS studies using larger datasets.

Reviewer : During GWAS, which correction was made?. There is no p-value of the GWAS nor threshold. Only the genomic intervals explaining 1% of the variance (based on a weak model) were taken. I believe that that methodology is not strong enough.

AU: We understand your concern. With regards to the model, please see our answer above. With regards to the single-step GWAS methodology, this is commonly used to perform GWAS in both human and livestock species (e.g., Wang et al., 2012; 2014; Irano et al., 2016; Lemos et al., 2016; Oliveira Silva et al., 2017; Marques et al., 2018; Silva et al., 2019; Grigoletto et al., 2019; 2020). The theory behind this method is explained in Wang et al. (2012; 2014).

References:               

Wang, H., Misztal, I., Aguilar, I., Legarra, A., & Muir, W. M. (2012). Genome-wide association mapping including phenotypes from relatives without genotypes. Genetics Research, 94(2), 73-83.

Wang, H., Misztal, I., Aguilar, I., Legarra, A., Fernando, R. L., Vitezica, Z., ... & Muir, W. M. (2014). Genome-wide association mapping including phenotypes from relatives without genotypes in a single-step (ssGWAS) for 6-week body weight in broiler chickens. Frontiers in Genetics, 5, 134.

Lemos, M. V., Chiaia, H. L. J., Berton, M. P., Feitosa, F. L., Aboujaoud, C., Camargo, G. M., ... & Mazalli, M. R. (2016). Genome-wide association between single nucleotide polymorphisms with beef fatty acid profile in Nellore cattle using the single step procedure. BMC genomics, 17(1), 213.

Medeiros de Oliveira Silva, R., Bonvino Stafuzza, N., de Oliveira Fragomeni, B., Miguel Ferreira de Camargo, G., Matos Ceacero, T., Noely dos Santos Gonçalves Cyrillo, J., ... & Misztal, I. (2017). Genome-wide association study for carcass traits in an experimental Nelore cattle population. PLoS One, 12(1), e0169860.

Marques, D. B., Bastiaansen, J. W., Broekhuijse, M. L., Lopes, M. S., Knol, E. F., Harlizius, B., ... & Lopes, P. S. (2018). Weighted single-step GWAS and gene network analysis reveal new candidate genes for semen traits in pigs. Genetics Selection Evolution, 50(1), 40.

Grigoletto, L., Brito, L. F., Mattos, E. C., Eler, J. P., Bussiman, F. O., Silva, B. D. C. A., ... & Ferraz, J. B. S. (2019). Genome-wide associations and detection of candidate genes for direct and maternal genetic effects influencing growth traits in the Montana Tropical® Composite population. Livestock Science, 229, 64-76.

Grigoletto, L., Ferraz, J., Oliveira, H. R., Eler, J. P., Bussiman, F. O., Abreu Silva, B. C., ... & Brito, L. F. (2020). Genetic Architecture of Carcass and Meat Quality Traits in Montana Tropical® Composite Beef Cattle. Frontiers in genetics, 11, 123.

Reviewer : The functional analysis methodology is not clear. Which genes were included in DAVID?. Only those observed in the overlapping area between the two GWAS??. How many genes were included in total Information is missing. The results expressed in p-values were corrected by any false discovery rate ?.

AU: This information has been clarified in the revised manuscript. The enrichment analysis was performed using the DAVID tool to select the GO terms and KEGG pathways with P≤0.05 and False Discovery Rate ≤ 10%. This information is presented in Table 4.

Reviewer : Additionally, the functional association with genes is weak from my point of view. For example, the MAPK pathway showed a 0.05 p-value. I suppose that it was higher to 0.05 (since only two digits were used), and therefore, it is NS, and cannot be used to support part of the discussion.

AU: We have added three digits in Table 4. The value was lower, but close to 0.05 (i.e. 0.049), so it is considered significant as we considered a P≤0.05 threshold. It is also worth pointing out that there are several recent discussions with regards to the interpretation of p-values. One could argue if there is actually biological difference between a p-value test of 0.049 and 0.051.

Reviewer : Finally, the fact that not there was no association with AMHR´s suggests that the variability among individuals in such receptors is low. That sounds weird to me (But it is an opinion)

AU: We were also surprised at first. However, this finding has been reported in other studies for the Nellore breed (e.g., Pierucci et al., 2019). We have discussed this point in the revised manuscript.  

Round 2

Reviewer 3 Report

Comments on the manuscript Genetic parameters and genome-wide association studies for anti- Müllerian hormone level and antral follicle population in Nellore cattle R1

Dear authors

Thanks for the clarification.

I believe that the manuscript is substantially improved by that.

The manuscript is quite clear in the present form and it reveals interesting data and results in an importing topic for the IVF industry. 

Most of my concerns were assessed in this new revised version.

However, I still have a major issue with the topic.

I strongly believe that your parameters and analysis are related to animal response to pharmacological treatment and not to the anti-mullerian and AFP per se.  

I understand that it is very challenging to assess both parameters without any standardization. But I hope the authors understand that it cannot change my concerns, nor it can be a factor that can be ignored. Moreover when it is quite demonstrated that the answer to a drug treatment it is not lineal nor equal among individuals.

Therefore, if you need to use a standardization protocol among individuals, you are estimating genetic parameters on the response to that protocol in both traits analyzed. 
I know that it is unknown, but maybe your results are different if you use a different treatment. Since the authors are not able to moderate, nor include in the model, such response, the conclusion cannot be extrapolated freely to the occurrence of both traits in natural conditions. 

I noticed that the authors included a phrase in the discussion and the conclusion assessing the issue. But I believe that it is not enough to be clear regarding the research topic.

Therefore, I suggest a change in the title and better clarification of the research topic across all the manuscript, assessing that parameters were estimated in two traits related to the response to the drug treatment. 

After that, the manuscript could be accepted for publication.

Author Response

Reviewer #3: “Dear authors,
Thanks for the clarification. I believe that the manuscript is substantially improved by that. The manuscript is quite clear in the present form and it reveals interesting data and results in an importing topic for the IVF industry. Most of my concerns were assessed in this new revised version. However, I still have a major issue with the topic. I strongly believe that your parameters and analysis are related to animal response to pharmacological treatment and not to the anti-Mullerian and AFP per se.

I understand that it is very challenging to assess both parameters without any standardization. But I hope the authors understand that it cannot change my concerns, nor it can be a factor that can be ignored. Moreover when it is quite demonstrated that the answer to a drug treatment it is not lineal nor equal among individuals. Therefore, if you need to use a standardization protocol among individuals, you are estimating genetic parameters on the response to that protocol in both traits analyzed.

I know that it is unknown, but maybe your results are different if you use a different treatment. Since the authors are not able to moderate, nor include in the model, such response, the conclusion cannot be extrapolated freely to the occurrence of both traits in natural conditions.

I noticed that the authors included a phrase in the discussion and the conclusion assessing the issue. But I believe that it is not enough to be clear regarding the research topic. Therefore, I suggest a change in the title and better clarification of the research topic across all the manuscript, assessing that parameters were
estimated in two traits related to the response to the drug treatment. After that, the manuscript could be accepted for publication.”

Authors: Thank you for the great feedback. We understand your concern and have modified the title and the manuscript accordingly. All changes are highlighted in the revised manuscript.
